# Pharmacist Workforce at Primary Care Clinics: A Nationwide Survey in Taiwan

**DOI:** 10.3390/healthcare9070863

**Published:** 2021-07-08

**Authors:** Wei-Ho Chen, Pei-Chen Lee, Shu-Chiung Chiang, Yuh-Lih Chang, Tzeng-Ji Chen, Li-Fang Chou, Shinn-Jang Hwang

**Affiliations:** 1Department of Medical Education, Taipei Veterans General Hospital, No. 201, Sec. 2, Shi-Pai Road, Taipei 11217, Taiwan; asdfg15995@gmail.com; 2Department of Pharmacy, Taipei Veterans General Hospital, No. 201, Sec. 2, Shi-Pai Road, Taipei 11217, Taiwan; kelseylee0612@gmail.com (P.-C.L.); ylchang@vghtpe.gov.tw (Y.-L.C.); 3Graduate Institute of Clinical Pharmacy, College of Medicine, National Taiwan University, Taipei 10617, Taiwan; 4Institute of Hospital and Health Care Administration, National Yang Ming Chiao Tung University, No. 155, Sec. 2, Linong Street, Taipei 11217, Taiwan; scchiang0g@gmail.com; 5Department of Family Medicine, Taipei Veterans General Hospital, No. 201, Sec. 2, Shi-Pai Road, Taipei 11217, Taiwan; sjhwang@vghtpe.gov.tw; 6Big Data Center, Department of Medical Research, Taipei Veterans General Hospital, No. 201, Sec. 2, Shi-Pai Road, Taipei 11217, Taiwan; 7Department of Public Finance, National Chengchi University, Taipei 116, Taiwan; lifang@nccu.edu.tw

**Keywords:** ambulatory care facilities, health workforce, pharmacists, Taiwan

## Abstract

Although dispensing is usually separated from prescribing in healthcare service delivery worldwide, primary care clinics in some countries can hire pharmacists to offer in-house dispensing or point-of-care dispensing for patients’ convenience. This study aimed to provide a general overview of pharmacists working at primary care clinics in Taiwan. Special attention was paid to clarifying the relationship by location, scale, and specialty of clinics. The data source was the Government’s open database in Taiwan. In our study, a total of 8688 pharmacists were hired in 6020 (52.1%) 11,546 clinics. The result revealed significant differences in the number of pharmacists at different specialty clinics among levels of urbanization. Group practices did not have a higher probability of hiring pharmacists than solo practices. There was a higher prevalence of pharmacists practicing in clinics of non surgery-related specialties than in surgery-related specialties. Although the strict separation policy of dispensing and prescribing has been implemented for 2 decades in Taiwan, most primary care clinics seem to circumvent the regulation by hiring pharmacists to maintain dominant roles in dispensing drugs and retaining the financial benefits from drugs. More in-depth analyses are required to study the impact on pharmacies and the quality of pharmaceutical care.

## 1. Introduction

Although the separation of prescribing and dispensing medication between physicians and pharmacists has been a common practice for a long time in North American and European countries, most Asian countries such as Korea, Malaysia, Japan, and Taiwan have only just begun to implement this separation system in recent decades [1]. Although the separation policy aimed to improve the quality of drug use, it could lead to patients’ inconvenience. Physicians in these Asian countries continuously struggled for the right to dispense medication. In Korea, revenue from drugs had once accounted for more than 40% of the total revenue in many clinics. To settle the strike by physicians about the implementation of a strict separation policy, the Korean Government raised physician fees by as much as 44%, thus adding an extra financial burden on patients [2]. In Malaysia, because the separation system encountered vehement opposition from physicians, separation of prescribing and dispensing only occurred in governmental healthcare facilities [3]. In Japan, the Government adopted a slight reform by increasing the reimbursement of prescription fees for facilities without in-house dispensing and decreasing fees for in-house dispensing. Additionally, clinics were allowed to hire pharmacists [4].

In Taiwan, because physicians resisted the separation system and lobbied for dispensing rights, the Government followed the policy in Japan. In 1997, they implemented a so-called “dual track system”, in which pharmacists could work either at independent pharmacies or at primary care clinics. The supporting argument was that since hospitals could hire pharmacists, clinics should be able to hire pharmacists in the same way [5]. According to the Pharmaceutical Affairs Act, any physician having dispensing facilities, could, for the purpose of medical treatment, dispense drugs by themselves based on their own prescriptions in remote areas where practicing pharmaceutical personnel was not available (as determined by the central or municipal competent health authorities) or in the case of urgent need of medical treatment services [6]. The strict separation policy restricted physicians from dispensing drugs independently in most cases. However, physicians could hire pharmacists to work for them at clinics. Therefore, the physician could still retain the benefits from dispensing fees and profit margins of medication. This may be contrary to the original goal of the separation policy, which expected the prescription given by the physician could be refilled at the community pharmacy. In Taiwan, the percentage of clinics hiring pharmacists in certain areas was more than 60% 1 year after the implementation of the new policy [7].

The retention of dispensing by clinics could be detrimental to the development of independent pharmacies. Although there was a wealth of literature on the introduction and influence of the separation system [1,2,3,8], there was a dearth of any published research on the pharmacist workforce at primary care clinics. The aim of this study was to conduct a nationwide survey of the pharmacist workforce at primary care clinics in Taiwan. Special attention was paid to geography, specialty, and scale of clinics. The unique phenomenon in Taiwan could offer valuable information for future discussion on healthcare policymaking.

## 2. Materials and Methods

### 2.1. Background

In Taiwan, the National Health Insurance program started in 1995 and covered almost all inhabitants [9]. There is no requirement for individuals to register with a primary care physician. Patients can freely consult with and switch between any kind of physician at local clinics and outpatient departments of hospitals without referral.

### 2.2. Data Source

Data were accessed through the website of Government’s open data in Taiwan (https://data.gov.tw/) (accessed on 1 December 2020) [10]. The basic characteristics of 359 townships in 23 cities and counties in Taiwan were collected from the Monthly Bulletin of Interior Statistics [11]. The Ministry of Health and Welfare provided data, including the number of clinics, physicians, pharmacies, and pharmacists in Taiwan [12,13].

### 2.3. Study Design

A descriptive, cross-sectional study of the nationwide pharmacist workforce at primary care clinics in 2016 was performed. The variables in this study, such as geographical conditions, the number of physicians per clinic, and physician practice types, might influence the clinic hiring pharmacists. Information about these variables was available from the Government’s open data. Therefore, we studied these factors to obtain deeper insight into current implementation of the policy for the separation of dispensing from prescribing in Taiwan.

To investigate the distribution of pharmacies and the pharmacist workforce in different regions, we adopted the urbanization stratification of Taiwan townships developed at Taiwan’s National Health Research Institutes [14]. The degree of urbanization of townships in Taiwan was determined by demographic characteristics such as population density, degree of industrialization, distribution of medical resources, number of physicians per 100,000 people, population ratio of farmers, people over 65 years old, and people with higher educational levels [14]. The 359 townships in Taiwan were stratified into seven levels of urbanization [14], and clinics were grouped into seven levels of urbanization according to their location. The seven levels of urbanization were introduced as follows. Level 1 townships, so-called highly urbanized townships, had highest population density, with people of highest educational levels, and highest medical resource density. Level 2 townships, so-called moderately urbanized townships, were second to level 1 townships in terms of population density, people with educational levels, and medical resource density. Level 3 townships (so-called emerging townships), and level 4 townships (so-called general townships) had medium levels of development. Level 5 townships, so-called aging townships, had highest proportion of the elderly and the lowest number of physicians per 100,000 people. Level 6 townships, so-called agricultural townships, had highest population ratio of farmers, the lowest population density, and people of lowest educational levels. Level 7 townships, so-called remote townships, had second-least number of physicians per 100,000 people. We defined urban areas as levels 1 and 2, suburban areas as levels 3 and 4, and rural areas as levels 5, 6, and 7. The number of clinics, pharmacies, and pharmacists were investigated according to their location at different levels. The percentage of clinics hiring pharmacists was calculated on the basis of the collected data.

Besides geography, other key variables of interest to this study were the number of physicians in a clinic and physician practice types.

The number of physicians per clinic indicated the scale of the clinic. The total number of physicians in a clinic was grouped as one, two, three, and ≥four. Then, we put all the data into a mosaic plot, with the horizontal axis showing the percentage of clinics with different numbers of physicians per clinic and the vertical axis showing the percentage of clinics with different numbers of pharmacists per clinic. A solo practice in our study was defined as a clinic with one physician. A group practice in our study was defined as a clinic with more than one physician (>1). 

Regarding physician specialty, although many physicians had more than one specialty certificate, they were categorized on the basis of the self-declared medical specialty as reported to the Government. In the present study, physician practice types were classified as a single-specialty practice or multi-specialty practice. We surveyed the pharmacist workforce in different single-specialty practices, including practices without specialist title, general medicine, family medicine, otolaryngology, pediatrics, ophthalmology, obstetrics and gynecology, dermatology, rehabilitation medicine, psychiatry, general surgery, plastic surgery, orthopedics, neurology, urology, neurosurgery, radiology, emergency, and anesthesiology. We grouped neurology, urology, neurosurgery, radiology, emergency, and anesthesiology as “others” because the number of clinics in these specialties was relatively small. We also analyzed multi-specialty practices, such as family medicine with pediatrics and family medicine with obstetrics and gynecology. The number of pharmacists hired by a clinic was classified into five groups: 0, 1, 2, 3, and ≥4. Then, we calculated the number of clinics in these groups. The percentage of clinics hiring pharmacists was calculated on the basis of the collected data. To perform additional statistical analysis by two-way ANOVA, we grouped specialty into six clusters according to their sample size and clinical pattern. We tried to keep the difference in sample size between groups as small as possible. In order to reduce intra-group variation, we performed grouping based on clinical practice patterns. The internal-medicine-related departments were grouped as group one. Surgery-related specialties were grouped as group two. Department of facial features (otolaryngology and ophthalmology) and pediatrics (patient’s condition being similar to those in otolaryngology) were clustered as group three. The remaining specialties were classified as group four. Due to the highest percentage of pharmacists at clinics and their specific specialty attributes, dermatology and psychiatry were independently grouped into group five and six, respectively.

### 2.4. Statistical Analysis

The mean and standard deviation were reported by urbanization level and specialty for all continuous variables. The percentage of group or solo practices hiring pharmacists was examined through chi-square tests. We stratified the data by urbanization level and specialty to illustrate their impact on clinics hiring pharmacists. The number of pharmacists at clinics was analyzed by two-way analysis of variance (ANOVA) with levels of urbanization and specialty as factors. All analyses were performed using the Statistical Package for Social Science (SPSS, version 23.0) with the significance level set at α = 0.05.

### 2.5. Ethical Approval

According to Taiwan’s personal data privacy legislation and the regulations of the institutional review board (IRB) at Taipei Veterans General Hospital (Taipei, Taiwan), the use of publicly available data was exempted from the IRB approval procedure.

## 3. Results

As the first step, 480 hospitals (e.g., academic medical centers, regional hospitals, and local hospitals) were excluded from the total of 22,936 nationwide medical institutions. As Chinese medicine was not covered by the separation policy and there were different regulations for dental clinics, we excluded Chinese medicine clinics (3996 clinics) and dental clinics (6873 clinics). Clinics in isolated isles such as Kingmen and Lienchiang counties (41 clinics) were excluded. Finally, a total of 11,546 clinics were included in our study.

### 3.1. Distribution of Pharmacies and Pharmacists Workforce in Urban, Suburban, and Rural Areas

Among 11,546 clinics in 359 townships in Taiwan, the majority was situated in urban areas (65.2%) and suburban areas (28.2%) (see Table 1). Similarly, 4587 (55.8%) pharmacies and 2944 (35.8%) pharmacies were in urban and suburban areas, respectively. Most pharmacists worked in urban areas (66.7%) and suburban areas (28.4%). In addition, the overwhelming majority of pharmacists in clinics were found in urban areas (66.8%) and suburban areas (28.0%).

### 3.2. Distribution of the Nationwide Pharmacist Workforce at Clinics, Stratified by Number of Physicians Per Clinic

Figure 1 shows the distribution of the pharmacist workforce in 11,454 clinics after excluding clinics without any physicians (92 clinics), stratifying the number of physicians per clinic into four groups. As for the clinics with pharmacists (orange, yellow, green, and blue areas in Figure 1), the percentage of clinics hiring three (green areas) and four or more (blue areas) per clinic became more when the scale of the clinics became larger (more physicians per clinic in other words). Regarding the clinics without any pharmacists (gray areas in Figure 1), less than half of clinics with one, two, and three physicians hired no pharmacists (48.0%, 46.2%, and 45.2%, respectively). Interestingly, more than half of clinics (55.6%) with four or more physicians did not hire pharmacists.

### 3.3. Distribution of Pharmacist Workforce in Different Specialties Clinics

Of 11,546 clinics in Taiwan, more than four-fifths of clinics (87.1%; 10,053/11,546) were single-specialty (see Table 2). Practices without a specialist title, general medicine, family medicine, otolaryngology, pediatrics, and ophthalmology accounted for three-fourths of the single-specialty clinics (76.5%; 7694/10,053), and practices without a specialist title were the largest proportion of clinics (29.1%; 2932/10,053). 

The percentage of clinics with pharmacists varied by specialty. The average percentage of single-specialty clinics with pharmacists was around half (51.7%; 5196/10,053). Among most single-specialty clinics, about two-thirds of those specializing in psychiatry (68.8%), ophthalmology (67.1%), and dermatology (66.4%) hired at least one pharmacist. Other specialties in which over half of the clinics hired pharmacists included pediatrics (65.1%), otolaryngology (61.0%), family medicine (54.0%), obstetrics, and gynecology (53.1%), and others (53.5%). One-third of general surgery clinics (35.5%) hired pharmacists, and very few rehabilitation medicine and plastic surgery clinics hired pharmacists (10.9% and 5%, respectively).

On average, around half of multi-specialty clinics hired pharmacists (55.3%; 825/1496). 

### 3.4. The Average Number of Pharmacists at Clinics by Urbanization Level and Specialty Group

To perform additional statistical analysis by two-way ANOVA, we grouped 15 specialties into six clusters according to their sample size and clinical pattern (see Table A1). Two-way ANOVA suggested that significant differences were observed in the number of pharmacists at different specialty clinics (F value of 5.8, degree of freedom = 5 and *p* value < 0.001) among levels of urbanization (F value of 2.3, degree of freedom = 6 and *p* value < 0.05). The results revealed an interaction between levels of urbanization and the specialty (F value of 2.2, and *p* value < 0.001). After we excluded group five and group six because of their smaller sample size, significant differences in the data still existed, along with the interaction between levels of urbanization and the specialty (F value of 2.8, and *p* value < 0.001).

The average number of pharmacists at clinics was put into a bar chart by urbanization level and specialty group (see Figure 2). Among specialty groups one to four, the average number of pharmacists at clinics in level 5 townships (red bar in Figure 2) was the lowest. Different colors of bars on the chart representing different levels of urbanization had a similar pattern, which revealed the lowest number of pharmacists at clinics in specialty group 2, and the highest one in specialty groups 5 and 6.

## 4. Discussion

To our knowledge, this was the first study to investigate the distribution of the pharmacist workforce at primary care clinics in Taiwan by location, scale, and specialty of the clinic.

It yielded several notable findings. First, there were significant differences in the number of pharmacists at different specialty clinics among levels of urbanization. Most clinics, pharmacies, and pharmacists were found in urban areas. Second, about half of clinics (52.1%) hired on-site pharmacists. We found that the larger the scale of the clinics, the higher the percentage of clinics that hired more than two pharmacists. The percentages of clinics hiring pharmacists were not obviously different between group practices versus solo practices. Finally, there was a lower probability of hiring pharmacists in surgery-related specialty clinics compared with non surgery-related clinics.

Globally, a lack of pharmacists in the workforce in rural areas has been reported in Australia, the United States, Canada, and Brazil [15,16,17,18,19,20]. Our study revealed an extremely low proportion of pharmacists working in rural areas (approximately 5%) in Taiwan. Based on our study, an uneven distribution of pharmacies was also found, with only 8.4% of pharmacies located in rural areas. The challenges for rural pharmacies and pharmacists’ practices were mainly based on economic realities [21]. The lack of pharmacists may increase pharmacy-related medication errors and alter the operations of the pharmacy department [22]. Besides, more part-time staff has been recruited, and the expanded use of overtime pay was noted [22]. One previous report indicated that patient safety, even death, could be contributed to by a shortage of pharmacists [23].

According to a previous study, more than 60% of clinics hired pharmacists in certain areas just after the new policy was launched [7]. After the new policy had been in place for more than 2 decades, our nationwide results were consistent with those of an earlier study. Based on our study, around half of clinics (52.1%) hired pharmacists. Such a high probability of clinics hiring pharmacists may be related to an oversupply of pharmacists and the risk of operating a pharmacy business. The ratio of physicians to practicing pharmacists was 1.4 to 1 in Taiwan, and the ratio of those was 3.9 to 1 in Organization for Economic Cooperation and Development (OECD) countries [24,25], indicating an oversupply of pharmacists relative to the number of physicians. For pharmacists, running their own pharmacy business was harder than being hired in clinics [26,27,28], so fewer pharmacists were willing to participate in the labor force in pharmacies.

Our study showed that the percentage of clinics hiring more than two pharmacists increased when the scale of clinics became larger (see Figure 1). As group practices could better build local healthcare market power compared to solo practices [29], we assumed that more primary care physicians in the clinics translated into more daily patient visits. To ensure the quality of dispensing, a threshold had been set at 80 prescriptions per day for each pharmacist. If there were more than 80 prescriptions, the dispensing fee was reduced by half [30]. Therefore, as the daily number of outpatient visits grew, we assumed that clinics would hire more pharmacists to work in shifts to handle the increasing patient demand for medical health services.

Based on our study, the percentage of clinics hiring pharmacists was not significantly different (*p* value of 0.41) for group versus solo practices (see Figure 1). Moreover, clinics with four or more physicians were the most likely to not hire pharmacists (55.6%) among all groups. This may be related to next-door pharmacies. To encourage prescriptions from clinics to be refilled in community pharmacies, the Government provided financial incentives to both clinics (prescription releasing fee) and pharmacies [31]. According to laws in Taiwan, pharmacies must be managed by pharmacists, but they can be owned by non-pharmacists [32]. Some practitioners found this loophole in the law, so they established pharmacies nearby and hired pharmacists to manage the pharmacy as their employees. These next-door pharmacies were controlled by practitioners (often physicians) and were distinct from independent pharmacies controlled by pharmacists [31]. Under this loophole in the law, the physicians benefited from both the prescription releasing fee and the pharmacist dispensing fee [33]. Previous research showed that large-volume clinics tended to collaborate with next-door pharmacies or contracted pharmacies with long-standing relationships [34]. In 2006, about one-third of pharmacies were next-door pharmacies [31]. Next-door pharmacies were not unique to Taiwan. They also occurred in Japan (the so-called “second pharmacy “) and the United States, but they gradually disappeared after governmental intervention from 1990 to 2000 [4]. In Taiwan, although the Government amended the law, the number of next-door pharmacies remains unknown.

When the number of patient visits or medication needs became greater, clinics were more likely to hire pharmacists to retain the financial benefits from drugs [7,35]. In our research about the distribution of the pharmacist workforce among different specialties, the results revealed a lower percentage of clinics with a pharmacist (<50%) in surgery-related specialties (including plastic surgery, general surgery, and orthopedics) than in most non surgery-related counterparts. The major medical service from clinics of rehabilitation medicine and surgical-related specialties seemed to be various therapies and surgical intervention, respectively [36,37,38]. Regarding many non surgical-related specialties, polypharmacy had been noted worldwide for decades [39,40,41]. For example, almost one-third of patients visiting outpatient psychiatry departments are on three or more psychotropic drugs in the United States [42].

Analysis results of a two-way ANOVA (see Table A1) suggested that significant differences were observed in the number of pharmacists at different specialty clinics (*p* value < 0.001) among levels of urbanization (*p* value < 0.05). The results revealed an interaction between levels of urbanization and the specialty (*p* value < 0.001). Among specialty groups one to four, our result (see Figure 2) showed the average number of pharmacists at clinics in level 5 townships was the lowest. According to the previous study, Level 5 townships had the lowest number of physicians per 100,000 people [14]. There seemed to be the lowest medical resource density in level 5 townships, which needs more attention in medical care. In different urbanization level townships (see Figure 2), the specialty group 2 (primarily surgery-related specialty) had the lowest number of pharmacists at clinics, and the specialty groups 5 and 6 (dermatology and psychiatry) had the highest one. In addition to urbanization level and specialty, clinics hiring pharmacists could be influenced by multiple factors such as clinic business type, the competitiveness of the market, the number of prescriptions refilled at a community pharmacy, physician’s trust in pharmacists, and practitioner’s beliefs being consistent with the core value of the separation policy.

Our comprehensive analyses of the nationwide distribution of the primary care pharmacist workforce by geographic location, the scale of the clinics, and different specialty types have some limitations. First, part-time doctors and pharmacists may cause imprecision in the calculation and presentation of data. Second, although we presume that clinics with four or more physicians collaborate with next-door pharmacies or long-term contracted pharmacies, the number of next-door pharmacies remains unknown at present [31]. Third, specialty clinics cannot be accurately counted because some clinics with multiple specialties may be considered as a single-specialty if they register only one specialty. Moreover, when seeking care in community clinics, many patients with illnesses are treated similarly by otolaryngologists, pediatricians, and family physicians [43]. Thus, there is an overlap of patients’ diseases among several specialties. Finally, we lacked information and could not consider the impact of potential confounding factors such as clinic business type, the competitiveness of the market, and the number of prescriptions refilled at a community pharmacy. We could obtain further information about the features of clinics with pharmacists if we implement a survey using a questionnaire. Hence, our study may not reflect a complete view of the pharmacist workforce in primary care clinics in Taiwan.

## 5. Conclusions

Our study shows significant differences in the number of pharmacists at different specialty clinics among levels of urbanization. Group practices do not have a higher probability of hiring pharmacists than solo practices. Clinics with non surgery-related specialties are more likely to hire pharmacists compared to surgery-related counterparts. In summary, a total of 8688 pharmacists have been hired to work in 6020 (52.1%) of 11,546 clinics, indicating that more than half of clinics hire on-site pharmacists. Although the strict separation between dispensing and prescribing has been implemented for 2 decades in Taiwan, most primary care clinics seem to circumvent the regulation by hiring pharmacists to maintain the dominant role in dispensing while maintaining control of the financial benefits from drugs. More in-depth analyses are required to further study the impact on pharmacies and the quality of pharmaceutical care.

## Figures and Tables

**Figure 1 healthcare-09-00863-f001:**
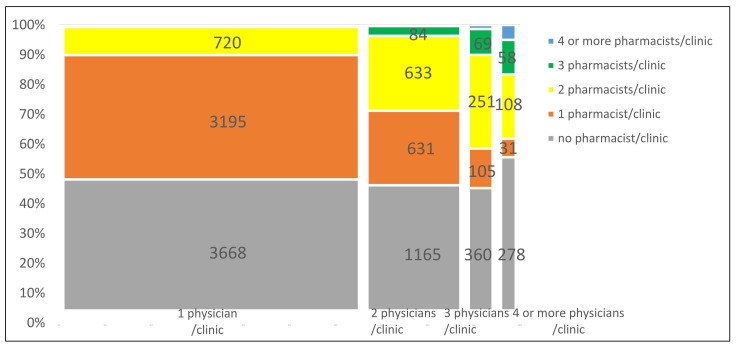
Distribution of the nationwide pharmacist workforce in clinics, stratified by the number of physicians per clinic. The numbers on the graph indicate the number of clinics.

**Figure 2 healthcare-09-00863-f002:**
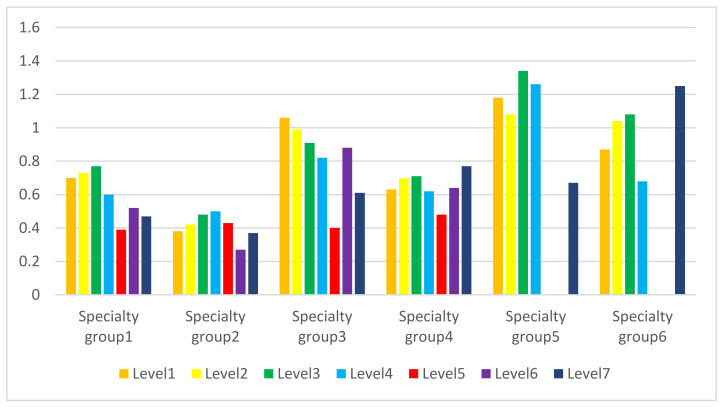
The average number of pharmacists at clinics by urbanization level and specialty group. We grouped specialty into six clusters according to their sample size and clinical pattern. The internal medicine-related departments were grouped as group one. Surgery-related specialties were grouped as group two. Department of facial features (otolaryngology and ophthalmology) and pediatrics (patient’s condition being similar to those in otolaryngology) were clustered as group three. The remaining specialties were classified as group four. Due to the highest percentage of pharmacists at clinics and their specific specialty attributes, dermatology and psychiatry were independently grouped into groups five and six, respectively. There were almost no clinics of specialty groups 5 or 6 in level 5 or 6 townships, so data were not applicable there.

**Table 1 healthcare-09-00863-t001:** Distribution of nationwide pharmacies and the pharmacist workforce at clinics in urban, suburban, and rural areas.

Urbanization Level	No. of Townships	No. of Pharmacists *	No. of Pharmacies	No. of Clinics	% of Clinics with Pharmacists	No. of Pharmacists in Clinics
Urban						
Level 1	27	10,872	1988	3526	50.5	2660
Level 2	43	12,659	2599	3998	53.4	3147
Suburban						
Level 3	56	5451	1696	1780	54.2	1424
Level 4	88	4561	1248	1478	51.0	1011
Rural						
Level 5	35	236	104	135	43.0	62
Level 6	61	661	266	295	51.9	162
Level 7	49	818	318	334	53.0	222
Total	359	35258	8219	11546	52.1	8688

***** The total number of pharmacists included those working in hospital, clinics, pharmacies, and the pharmaceutical industry.

**Table 2 healthcare-09-00863-t002:** Distribution of pharmacist workforce in clinics, stratified by specialty.

Specialty	Number of Clinics						% of Clinics with Pharmacists
	0 Pharmacist/Clinic	1 Pharmacists/Clinic	2 Pharmacists/Clinic	3 Pharmacists/Clinic	≥4 Pharmacists/Clinic	Total	
Single-specialty clinics	4857	3445	1504	212	35	10,053	51.7
Practices without specialist title	1558	1057	282	32	3	2932	46.9
General medicine	566	371	133	11	0	1081	47.6
Family medicine	493	391	158	27	3	1072	54.0
Otolaryngology	388	298	263	42	5	996	61.0
Pediatrics	317	344	211	31	5	908	65.1
Ophthalmology	232	286	164	20	3	705	67.1
Obstetrics and gynecology	251	218	62	2	2	535	53.1
Dermatology	150	133	126	26	12	447	66.4
Rehabilitation	278	31	3	0	0	312	10.9
Psychiatry	88	132	50	11	1	282	68.8
General surgery	160	74	13	0	1	248	35.5
Plastic surgery	209	11	0	0	0	220	5.0
Orthopedics	114	55	25	7	0	201	43.3
Others	53	44	14	3	0	114	53.5
Multi-specialty clinics	668	545	217	50	13	1493	55.3
All clinics	5525	3990	1721	262	48	11,546	52.1

## Data Availability

Not applicable.

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
