# Peer review of "Pharmacist Workforce at Primary Care Clinics: A Nationwide Survey in Taiwan"

_healthcare, 2021, doi:10.3390/healthcare9070863_

Round 1
Reviewer 1 Report
The theme of the research on distribution of pharmacists in primary care settings was interesting. However, there are several points that require consideration.
- Background
The significance of objective of the study seemed vague, especially for readers who are not familiar with situation of pharmacists in Taiwan. It was difficult to understand the author’s implication in the sentence from L60. Could you describe more specifically the expected effects of achieving this research objective?
- Methods
Was there any reason for setting geographical conditions, the number of pharmacists per clinic, and physician practice types as variables in this study?
- Results
Please clarify why Chinese medicine clinics were excluded from the study.
- L147: The results on level 3 towns and level 5 towns were difficult to interpret. It may have been because difference between level 3 and 5 were not clearly described other than that level 3 was categorized as suburban and level 5 was rural.
- The results of the chi-square test seem to be shown only in the legends of Table 1. It may be difficult for readers to access to the results.
- The results described from L155 were difficult to read because of too many fractional descriptions. Please clarify the description of the results by properly using fractions and percentages.
- Discussion
The discussion part is redundant. Please clarify logically what the authors want to discuss from the results of this study.
Author Response
"Please see the attachment."

Reviewer 2 Report
Thank you for giving me the opportunity to peer review this manuscript, which presents a descriptive analysis of pharmacist workforce at primary care clinics in Taiwan. Overall the manuscript is well structured and clearly written, and the topic is important for primary care medical and pharmacy practice in Taiwan. However there are some issues related to the analysis and interpretation of the percentages of clinics hiring pharmacists based on urbanisation levels of townships. As a result, the conclusion of urban-rural disparity (in terms of clinics hiring pharmacists) does not seem to be supported by the data and the authors may wish to re-visit this. More detailed comments are provided below.
Major comments
- The data and analyses presented by the authors demonstrated a clear disparity in terms of the absolute number of pharmacists (and pharmacies and clinics) between urban, suburban and rural areas. However the analysis to examine the urban–rural disparity in the proportion of clinics hiring pharmacists and the interpretations of the finding of this analysis require some clarification. Based on the description in the Methods section, the seven levels of urbanisation were grouped into three categories of urban, suburban and rural areas/townships. However, the data presented for the chi squared test (X2(6) = 14.8, p= 0.022) under Table 1 showed a degree of freedom of 6, which suggested that the analysis was carried out based on the original seven urbanisation levels rather than the three urbanisation categories (which would result in a degree of freedom of 2). Looking at the data in Table 1, it seems the disparity isn’t necessarily an urban-rural divide; the percentage of clinics with pharmacists appears to be particularly low in level 5 townships, but the percentages for levels 6 and 7 seem to be comparable to those of urban and suburban areas (levels 1-4). Therefore using the original seven levels for chi squared test might not be a bad idea. However, describing (and interpreting/concluding) the statistically significant result for the chi squared test as an urban-rural divide does not seem to accurately reflect the data, and these might need to be re-considered (e.g. do the data highlight a unique difference for level 5 townships rather than an urban-rural divide? Were the differences in the percentages of clinics hiring pharmacists confounded by speciality, e.g. did level 5 township have a higher proportion of surgery related clinics, or did level 6 and 7 township have a lower proportion of surgery related clinics?)
- Lines 239-241: the ‘post hoc’ analysis described here does not seem to be particularly helpful, e.g. why only level 1 and level 5 were compared?
- Lines 247-250: the arguments offered here actually explained/supported why there might not be urban-rural divide in terms of percentages of clinics hiring pharmacists?
- Table 2 needs some clarification. Were ‘practices without speciality title’ a sub-category of ‘single-specialty clinics’? Based on line 114, this seems to be the case. If so, it might be helpful to present Table 2 with row headings left-aligned, with sub-categories (including ‘practices without speciality title) under ‘Single-speciality clinics’ indented to make it clear that they are sub-categories of ‘Single-speciality clinics’.
- Line 196: the difference seems to be relatively small – is it statistically significant / practically important?
Minor comments
- Line 53: ‘dual track system’ may be a better description than ‘double track system’?
- Line 76: It may be more precise to say ‘There is no requirement for individuals to register with a primary care physician’?
- Line 84: in each ‘city and county’ in Taiwan? Please clarify the relationship between the 359 townships mentioned above and the cities and counties, e.g. The xx cities and counties covered xx% of / all 359 townships in Taiwan.
- Lines 91, 141, 147 & 148: should it be ‘townships’ rather than ‘towns’ in these lines? The former tends to be used to describe an area with a local administration whereas the latter usually refers to a built-up area larger than a village but smaller than a city, and therefore they are not synonyms.
- Line 135: ‘As the first step’ might be better than ‘Initially’.
- Line 161: should it be three or more ‘physicians’ rather than ‘pharmacists’, as all the preceding sentences talked about the number of physicians rather than pharmacists?
- Lines 269-270: maybe better to say ‘a threshold has been set at 80 prescriptions per day for each pharmacist’?
- Line 417: ‘when seeking care’ rather than ‘’when practicing’?
- Table 1: could the authors provide a brief description for each of the seven levels (either in the Methods section or as a footnote for the table) to facilitate readers’ understanding of the context? (based on my understanding – level 1: highly urbanised township: 2: moderately urbanised township, 3: emerging township, 4: general township, 5: ageing township, 6: agricultural township, 7: remote township?)
Author Response
"Please see the attachment."

Round 2
Reviewer 1 Report
Dear Authors,
Thank you for your revised manuscript.
Although the manuscript is well revised, there were several points of my concern.
- Methods
On page 6, it was unclear why the authors grouped the clinics into six specialty groups. Were they based on some kind of categories? Please clarify.
- Figure 2
It may be better to show the results using bar graph instead of line graph.
- Discussion
I am still very concerned about the redundancy of the discussion. The authors explained that the main text is required to be around 3000 words, but I found that the current manuscript has around 4500 words. Too long discussion makes the aim of the manuscript unclear. Please reconfirm that the background/aim and discussion/conclusion correspond.
Reviewer 2 Report
Many thanks for giving me the opportunity to review the revised manuscript. I appreciate the authors’ careful consideration of my comments. The changes made have satisfactorily addressed all my previous comments.
I only have one minor query and some suggestions related to wording in various places of the manuscript. These are described below.
1. Lines 400-401: the ‘ratio’ may need further explanation. If the ‘prescription refilled at pharmacy’ is numerator, what is the denominator? Are you referring to independent community pharmacy here, or any pharmacy?
2. Some suggestions concerning the choice of words:
Line 94: ‘obtain’ rather than ‘get’
Line 95: would ‘current implementation of the policy for the separation of dispensing from prescribing in Taiwan’ clearer than ‘the formation of the current separation system in Taiwan’?.
Lines 104-114; 382-384: the word ‘got’ is rather colloquial and is not commonly used in academic writing; the authors may wish to use ‘had/has/have’ instead.
Line 111: ‘lowest’ rather than ‘minimum’
Lines 146-147: The mean and standard deviation were reported ‘by’ urbanization level and specialty ‘for’ all continuous variables.
Line 160-161: “because Chinese medicine was not covered by the separation policy and there were different regulations for dental clinics.”
Lines 181-183: “the percentages of clinics hiring three (green areas) or four or more (blue areas) pharmacists per clinic became greater when the scale of the clinic became larger (more physicians per clinic in other words).”
Line 200: “Other specialities in which over half of the clinics hired pharmacists included pediatrics (65.1%), otolaryngology (61.0%) ….”
Line 202: ‘clinic’ rather than ‘Clinic’
Line 333: “Similarly, a relatively low percentage…”
Lines 382-382: The specialty group 3 (“primarily” surgery-related specialty) “had” the lowest number of pharmacists at clinics
Line 387: ‘belief’ rather than ‘idea’
Lines 389-391: “…nationwide distribution of the primary care pharmacist workforce by geographic location, scale of the clinics, and different specialty types have some limitations.”
Lines 391-392: “..may cause imprecision in the calculation and presentation of data”.
Lines 399-400: “Finally , we lacked information and could not consider the impact of potential confounding factors such as clinic business type, competitiveness of the market, and ratio of prescription refilled at pharmacy.”
Line 401: ‘obtain’ rather than ‘get’
Line 412: “strict separation between dispensing and prescribing”
